# Factors associated with the use of dental services in older adults in Peru

**Bryan Alexis Cossio-Alva**[1]*, **Rubén Espinoza Rojas**[2], **Miguel Angel Ruiz-Barrueto**[1], **Giancarlo Becerra Atoche**[1], **Christian R. Mejia**[3], **Ibraín Enrique Corrales-Reyes**[4]

1 Facultad de Ciencias de la Salud, Escuela de Estomatología, Universidad César Vallejo, Piura, Peru, 2 Instituto de Investigaciones en Ciencias Biomédicas de la Universidad Ricardo Palma, Peru, 3 Asociación Médica de Investigación y Servicios en Salud, Lima, Peru, 4 Manatee Oral and Facial Surgery Center, Bradenton, Florida, United States of America

* bcossio@ucv.edu.pe

## Abstract

### Background

The use of dental services by older adults in Peru faces various challenges that impact both their oral and overall health. Several factors play a crucial role in obtaining adequate dental care.

### Aims

To evaluate the factors associated with the use of dental services in older adults in Peru.

### Methods

This study was an observational, analytical, and cross-sectional investigation that utilized data from multiple years (2018–2022) of the Demographic and Family Health Survey (ENDES, by its Spanish acronym). The use of dental services was assessed through a specific survey question and analyzed in association with various socio-demographic variables, employing both descriptive and analytical statistical methods.

### Results

In the multivariable analysis, the likelihood of not utilizing dental services was higher among men aPR: 1.53 (95% CI: 1.45–1.61), those without health insurance aPR: 1.44 (95% CI: 1.36–1.53), those who self-identified as other races aPR: 1.22 (95% CI: 1.13–1.33), Afro-descendants aPR: 1.10 (95% CI: 1.01–1.19), whites aPR: 1.12 (95% CI: 1.01–1.25) according to education level, those with a physical limitation aPR: 1.24 (95% CI: 1.15–1.33), and increased as they were poorer; it was lower among the rich but more pronounced among the middle-income, poor, and very poor, adjusted for five variables.

### Conclusion

It was reported that 15% of older adults did not use dental services, and this was associated with significant socio-demographic variables.

**Data Availability Statement:** The data have been deposited at the following DOI: 10.6084/m9.figshare.27969189.

**Funding:** This study was funded by Universidad César Vallejo (N° P-2024-131-VI-UCV).

**Competing interests:** The authors have declared that no competing interests exist.

## Introduction

The state of oral health is closely related to the general state of health of individuals [1, 2]. The various affections in the stomatognathic system adversely compromise living conditions, which in turn make it difficult to grind food, communication, interaction with others, as well as poor self-perception of the aesthetic component and various psychosocial imbalances [3, 4]. Additionally, poor oral health is considered a risk factor for coronary heart disease, stroke, dental caries, trauma, hereditary conditions and periodontopathies. All of these are increased by inadequate oral hygiene, tobacco consumption, poor diet, immunodeficiencies, advanced age, etc [5].

The World Health Organization (WHO) [6] indicates that the population group of people aged 60 years and older is growing at a faster rate than that of other age groups. Furthermore, in 3 decades, 80% of these older adults will live in developing countries. This trend is also shown in Peru, where it is estimated that by 2050, there will be almost 9 million Peruvians in this group [7]. This is relevant because older adults experience physiological changes and they are also more vulnerable to the challenges of society. Likewise, oral diseases are highly prevalent, as they are linked to the presence of systemic diseases, increased consumption of drugs, physical weakness and lack of access to dental services [8]. This highlights the importance of oral health in general and how its deterioration has negative implications on overall health and quality of life [9]. Globally, access to dental services is influenced by various related factors ranging from socioeconomic to cultural and geographic aspects. In this regard, it is important to consider variables such as income level, education, availability of services, among others [10]. This is why there is a high prevalence of missing teeth, dental caries, periodontal diseases, temporomandibular disorders, infections, among others [3, 11].

In this context, it is extremely important to direct the necessary resources in order to use them in a timely manner and thus improve the quality of dental services, as well as to train the professionals responsible for the area. Borg-Bartolo et al. [12] found that dental caries and tooth loss are very frequent and present differences according to socioeconomic level, age groups and countries. On the other hand, Northridge et al. [13] point out that people with low income, without insurance or belonging to minority groups are more likely to have poor oral health, and that the main barriers to receiving care in health services are not having insurance, high prices, perception of the need for treatment, among other factors [14].

In the United States, access to dental services for older adults is limited and this is reflected in the high prevalence of oral diseases, due to the fact that certain state health insurances do not cover dental care. The high cost of treatment in private facilities and poor access to these services in state insurance policies affect the oral health of the population [15]. On the other hand, in Germany, there was a high frequency of use of dental services by older adults during the COVID-19 pandemic [16].

In Brazil, a low use of dental services by older adults from a minority community was identified, with differences existing according to ethnicity and individual perception of oral health [17]. In Peru, Azañedo et al. [18] identified in 2018 a low use of dental services by older adults and the factors that were associated were area of residence, educational level, health insurance, geographic domain and welfare quintiles.

This study provides a more recent and longitudinal analysis by taking data from multiple years of ENDES. Unlike previous research, that relied on single-year data or addressed only a limited set of factors, this investigation integrates information collected over an extended period, allowing for the identification of trends and consistent patterns in the utilization of dental services among older adults. Furthermore, the study stands out by examining specific socio-demographic variables that have been underexplored in Peruvian contexts, such as self-

perceived ethnicity, physical limitations, and wealth index, offering a more comprehensive perspective on the barriers and facilitators of dental care access. This innovative approach not only enhances the understanding of oral health inequalities but also provides robust evidence for the development of more inclusive and effective public policies in oral health care in Peru [14–16, 18, 19]. Therefore, the objective was to evaluate the factors associated with access to dental services in older adults in Peru.

## Methods

An observational, analytical and cross-sectional research was conducted, based on the ENDES initiative occurring in Peru. This survey employs a probabilistic, stratified, and multi-stage sampling design, ensuring the representativeness of results at the national level, as well as for rural and urban areas and across the country's various regions. Data are collected annually through interviews conducted in selected households, following standardized protocols that guarantee the quality and comparability of the information. Notably, ENDES gathers repeated cross-sectional data, meaning that each year new samples are selected without links to previous years, thereby preserving participant anonymity. This approach enables temporal analyses and the evaluation of health indicator trends while adhering to ethical principles of confidentiality and data protection.

The data was obtained in SPSS format, through the website of the National Institute of Statistics and Informatics (INEI, from the Spanish initials) (https://proyectos.inei.gob.pe/endes/). The data quality control process for this study was conducted systematically to ensure the consistency, accuracy, and reliability of the information obtained from ENDES. First, internal consistency checks were performed by cross-referencing key variables to identify and correct potential discrepancies, such as inconsistencies in age, gender, and other sociodemographic characteristics. Variable distributions were also analyzed to detect outliers that could distort the results. These data points were reviewed and excluded from the analysis if deemed irrelevant or implausible. Missing data were addressed using statistical imputation techniques for key variables, while variables with substantial missing data that could not be reliably imputed were excluded from the multivariate analysis to preserve model validity. Subsequently, the data were consolidated into a single database after conducting additional reviews to ensure uniformity across the different years of the study. Finally, a pilot exploratory analysis was conducted to identify potential residual issues such as collinearity or inconsistencies, ensuring that the dataset was in optimal condition for analysis. This meticulous process provided robust and reliable results, minimizing the potential impact of errors inherent to the use of secondary data.

The research was approved by the Research Ethics Committee of the School of Stomatology of the Universidad César Vallejo (Registration Code: N˚049-2023-/UCV/P). By having access only to the coding of the participants, the identity of each participant was protected, respecting the principle of confidentiality.

The universe and the sample were identical, consisting of all older adults surveyed (60 years of age and older) (N = 30318) in the study period. The ENDES modules, where the variables to be studied were identified: household characteristics, basic data on women of reproductive age and the health survey.

### Outcome

The variables considered for evaluating the associated factors were selected based on the evidence from current background studies [2, 4, 5, 8, 12–16, 18, 19]. The primary outcome of this study was the utilization of dental services, defined dichotomously by categorizing participants

into two groups: those who had used dental services within the past two years and those who had not. This definition was based on a survey question regarding the last dental visit, allowing for a distinction between recent and non-recent utilization. The two-year period was selected considering the absence of specific national guidelines in Peru regarding recommended dental visit frequency, providing a reasonable range for capturing recent utilization compared to international standards. Moreover, this approach aligns with previous studies in similar contexts, facilitating the comparability of findings. This criterion also reflects the characteristics of the Peruvian healthcare system, where utilization may be influenced by geographic, economic, and cultural barriers, making shorter intervals less representative of the national reality.

## Predictors

Relevant sociodemographic predictors were considered, including sex (male or female), age (60–69, 70–79, 80–89, or 90 and older), school attendance (yes or no), and health insurance coverage (yes or no).

## Covariates for adjustment

Adjustment covariates included native language (Quechua, Aymara, Awajún/Aguaruna, Spanish, foreign language or others), ethnoracial self-perception (Quechua, Afro-descendant, White, Mestizo or others), education (none, primary school, secondary school or higher), geographical region (Amazonas, Ancash, Apurimac, Arequipa, Ayacucho, Cajamarca, Callao, Cusco, Huancavelica, Huánuco, Ica, Junín, La Libertad, Lambayeque, Lima, Loreto, Madre de Dios, Moquegua, Pasco, Piura, Puno, San Martín, Tacna, Tumbes or Ucayali), region you live in (metropolitan Lima, rest of the coast, mountains or jungle), type of residence (urban or rural), marital status (single, married/cohabiting or widowed/separated), physical limitation (yes or no), wealth index (poorest, poor, middle, rich, richest), considering the possession of assets, housing characteristics, and access to basic services, providing a comprehensive measure of socioeconomic conditions beyond direct income.

## Statistical analysis

The statistical analysis was carried out using Stata software (version 18), performing an exhaustive analysis of the qualitative variables through frequency tables and percentages. To investigate the relationships among these categorical variables, the Chi-square statistic was applied, with Yates´s correction.

In evaluating the factors associated with the utilization of dental services in older adults, the crude prevalence ratio (cPR) was calculated along with its corresponding confidence intervals and p-values. Subsequently, the adjusted prevalence ratio (aPR) was determined using a modified Poisson regression with robust standard error estimates to account for overdispersion in the data.

The main criterion for a variable to move from the crude models to the adjusted one was a p-value of 0.05 or less; except for the variable region where the person lives (which was not included due to collinearity with other variables). All this with a confidence level of 95% and a statistical significance set at $p < 0.05$.

## Results

Of the 30,318 respondents, 15.0% (4,540) did not utilize dental services. Among the participants, 52.7% (15,988) were women, the most common age range was 60–69 years (53.5%), 86.8% had attended school, and 84.8% were registered for health insurance, 73.1% had Spanish

as their native language, 44.6% perceived themselves as Mestizo, 38.9% had attained primary education, 37.2% resided in Metropolitan Lima, 77.9% living in an urban area, 53.9% were married/cohabiting, 8.3% reported having a physical limitation, and 25.3% reported being in the "richest" group according to their wealth index.

When an initial cross-check of dental service use against other variables was conducted, statistically significant differences were observed based on sex (p<0.001), having health insurance (p<0.001), ethnoracial self-perception (p<0.001), educational level (p<0.001), the region which you live in (p<0.001), type of residence (p<0.001), whether you had physical limitation (p<0.001), and wealth index (p<0.001) **Table 1.**

When the percentage of dental service use was analyzed by department, the lowest rates were observed in Cajamarca (60%), Piura (64%), and Loreto and Puno (67% each). In contrast, the highest rates were found in Tacna (95%), and in Lima and Callao (94% each), with a statistically significant difference identified (p-value <0.001, based on the chi-square test with Yates's correction) **Fig 1.**

When multivariable analysis was performed, it was found that the likelihood of not using dental services was higher among men (aPR: 1.53; CI 95%: 1.45–1.61), those who did not have health insurance (aPR: 1.44; CI 95%: 1.36–1.53), those who self-perceived to be of other races (aPR: 1.22; CI 95%: 1.13–1.33), people of African descent (aPR: 1.10; CI 95%: 1.01–1.19), among whites (aPR: 1.12; CI 95%: 1.01–1.25), according to educational level, those who had a physical limitation (aPR: 1.24; CI 95%: 1.15–1.33) and increased as one was poorer, among the rich it was lower (aPR: 1.22; CI 95%: 1.05–1.43), but was more accentuated in those with middle income (aPR: 1.91; CI 95%: 1.65–2.21), the poor (aPR: 2.55; CI 95%: 2.19–2.96) and among the very poor (aPR: 3.77; CI 95%: 3.22–4.41), adjusted for educational level, type of residence, department of residence and native language **Table 2.**

## Discussion

Although oral diseases affect people of all ages, the geriatric population is one of those that show greater deterioration in their oral health, mainly due to existing chronic diseases associated with aging [20, 21]. In Peru, even though the state identified oral health as a health problem, it was ultimately not included in the list of national health research priorities for 2019–2023 [22]. Additionally, it has been consistently reported that as older adults get old, there is a decrease in the use of dental services, which results in poorer oral health [23].

In the present investigation, it was reported that one in six respondents did not have access to dental services during the study period. Men accessed dental health services to a lesser extent, similar to what was reported by Gaskin et al. [19] in the U.S., who found that men were less likely to visit the dentist in the last 5 years. Similarly, Galvão et al. [24] in Brazil associated no dental visits with being male. Consequently, Corovic et al. [25] in Serbia established that men were approximately 1.8 times more likely than women to not use dental services.

We also found that lacking health insurance was strongly associated with not being able to use dental care, which aligns with findings published by Gaskin et al. [19] who reported that health insurance status was an important predictor of edentulism in older adults. Those living in areas with better health insurance coverage were more likely to preserve their teeth and visit the dentist at least once in the last 5 years. Similarly, Marchini et al. [26] reported that in the United States, about two-thirds of older adults do not have dental insurance, making access to dental care less likely [27]. In relation to this, Jang and colleagues [28] reported that 71% of Korean-American individuals over 60 years old lacked dental health insurance, and 38% of them did not have any dental visits in the last year. Similarly, Galvão et al. [24] and Hartmann et al. [29] established that one of the factors associated with older adults having their last dental

**Table 1. Description of factors associated with the use of dental services among older adults in Peru.**

| Variables | Access to dental services | | | p-Value |
|---|---|---|---|---|
| | N total | Yes n (%) | No n (%) | |
| **Sex** | | | | |
| Female | 15988 | 13897 (86.9) | 2092 (13.1) | <0.001 |
| Male | 14329 | 11881 (82.9) | 2448 (17.1) | |
| **Age (years)** | | | | |
| 60–69 | 16208 | 13816 (85.2) | 2392 (14.8) | 0.422 |
| 70–79 | 9057 | 7700 (85.0) | 1357 (15.0) | |
| 80–89 | 4387 | 3705 (84.4) | 683 (15.6) | |
| 90 or Older | 666 | 557 (83.7) | 109 (16.3) | |
| **Health insurance** | | | | |
| Yes | 25701 | 22104 (86.0) | 3597 (14.0) | <0.001 |
| No | 4616 | 3673 (79.6) | 943 (20.4) | |
| **Ethnoracial self-perception** | | | | |
| Quechua | 8119 | 6804 (83.8) | 1314 (16.2) | <0.001 |
| Others | 3812 | 2900 (76.1) | 912 (23.9) | |
| Afrodescendant | 2856 | 2169 (75.9) | 687 (24.1) | |
| White | 2000 | 1714 (85.7) | 286 (14.3) | |
| Mestizo | 13531 | 12190 (90.1) | 1341 (9.9) | |
| **Education** | | | | |
| None | 4241 | 2805 (66.1) | 1437 (33.9) | <0.001 |
| Primary School | 11802 | 9578 (81.2) | 2224 (18.8) | |
| High School | 7972 | 7289 (91.4) | 683 (8.6) | |
| Higher | 6302 | 6106 (96.9) | 196 (3.1) | |
| **Region living in** | | | | |
| Lima (capital) | 11271 | 10586 (93.9) | 686 (6.1) | <0.001 |
| Coast | 7811 | 6627 (84.8) | 1185 (15.2) | |
| Mountains | 8165 | 6319 (77.4) | 1846 (22.6) | |
| Jungle | 3070 | 2246 (73.2) | 824 (26.8) | |
| **Type of residence** | | | | |
| Urban | 23614 | 21303 (90.2) | 2311 (9.8) | <0.001 |
| Rural | 6703 | 4474 (66.7) | 2229 (33.3) | |
| **Marital Status** | | | | |
| Single | 1777 | 1501 (84.5) | 276 (15.5) | 0.063 |
| Married/cohabiting | 16327 | 13955 (85.5) | 2372 (14.5) | |
| Widowed/separated | 12214 | 10322 (84.5) | 1892 (15.5) | |
| **Physical Limitation** | | | | |
| No | 27815 | 23839 (85.7) | 3975 (14.3) | <0.001 |
| Yes | 2503 | 1939 (77.4) | 565 (22.6) | |
| **Wealth Index** | | | | |
| Poorest | 6578 | 4194 (63.8) | 2384 (36.2) | <0.001 |
| Poor | 4644 | 3730 (80.3) | 913 (19.7) | |
| Middle | 5410 | 4794 (88.6) | 616 (11.4) | |
| Rich | 6028 | 5669 (94.0) | 359 (6.0) | |
| Richest | 7658 | 7390 (96.5) | 267 (3.5) | |

P values obtained with the chi-square test, with Yates´s correction.

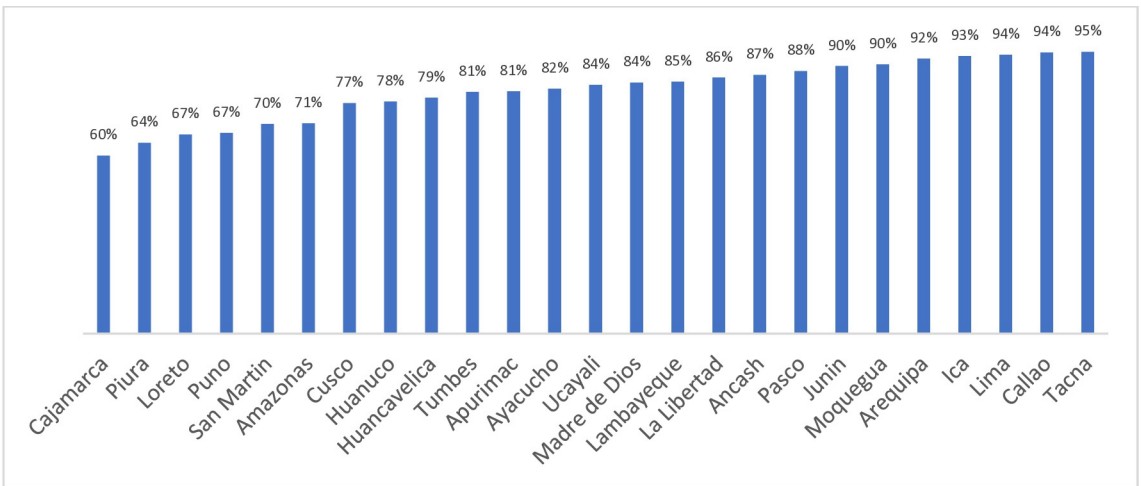

**Fig 1. Departmental percentages of dental service use among older adults in Peru.**

visit in the last 12 to 36 months or delaying seeking oral health services was the lack of medical or private dental insurance. Additionally, Qu et al. [30] as well as Xu et al. [31] reported that 41% of older adults without medical insurance had never received dental care, and this largely depended on having basic or government health insurance.

Additionally, it was found that self-identifying with certain specific races or ethnicities was associated with lower use of dental services, particularly among Afro-descendants and whites. These data align with those reported by Mao et al. [32] who found that 77% of elderly Chinese Americans did not have access to dental care. According to Cha et al. [33], race is a predictive factor associated with delays in dental care. This was corroborated by Bhoopathi et al. [34], who found that although non-Hispanic Black Americans and Hispanic adults were more likely than non-Hispanic whites to report an unmet need for dental care, the association between Hispanics and non-Hispanic whites was statistically significant. Kramarow [23] reported that 69% of non-Hispanic white seniors over the age of 65 were more likely to have had a dental visit in the last 12 months, compared to 55% of Hispanics, 53% of non-Hispanic Blacks, and 53% of non-Hispanic Asians. According to Wu et al. [35] racial and ethnic disparities in the use of dental services among Whites, Hispanics, Blacks, Asians, American Indians or Alaska Natives, and Native Hawaiians or other Pacific Islanders increased with age. Additionally, older Black adults had the lowest level of dental service utilization (65%), compared to the two highest groups: older white adults (79%) and older Asian adults (76%). Similarly, Liu et al. [36] reported that Black participants in Tennessee were less likely to report dental visits. Furthermore, race and ethnicity were not only associated with lower use of dental care but also served as strong predictors of edentulism and unmet dental care needs [19]. In this context, da Silva-Sobrinho et al. [17] found that self-identified Afro-descendant elderly individuals of mixed race in Brazil had reduced access to dental services. Additionally, Chan et al. reported that race influenced delays in both medical and dental care for elderly individuals during the COVID-19 pandemic, as reported by Chan et al. [37], who found that non-Hispanic whites and those of other racial/ethnic origins experienced more delays in care than non-Hispanic Blacks.

Likewise, it was found that a lower level of education or lack of formal schooling was associated with a lower frequency of dental service use. This is correlated with what was reported by Andrade et al. [38] who, in their study of 23 countries, revealed that the likelihood of accessing dental services increased among elderly individuals with higher educational levels. Elderly

**Table 2. Bivariate and multivariable analysis of factors associated with the lack of utilization of dental services among older adults in Peru.**

| Variables | Crude analysis | | Adjusted analysis | |
|---|---|---|---|---|
| | cPR (95% CI) | p value | aPR (95% CI) | p value |
| **Sex** | | | | |
| Female | Category of comparison | | Category of comparison | |
| Male | 1.31 (1.24–1.38) | <0.001 | 1.53 (1.45–1.61) | <0.001 |
| **Age (years)** | | | | |
| 60–69 | Category of comparison | | Did not enter the model | |
| 70–79 | 1.01 (0.94–1.07) | 0.870 | Did not enter the model | |
| 80–89 | 1.02 (0.94–1.11) | 0.602 | Did not enter the model | |
| 90 or Older | 1.05 (0.88–1.27) | 0.579 | Did not enter the model | |
| **Health insurance** | | | | |
| Yes | Category of comparison | | Category of comparison | |
| No | 1.47 (1.38–1.57) | <0.001 | 1.44 (1.36–1.53) | <0.001 |
| **Ethnoracial self-perception** | | | | |
| Mestizo | Category of comparison | | Category of comparison | |
| Quechua | 1.56 (1.45–1.68) | <0.001 | 1.01 (0.91–1.13) | 0.806 |
| Others | 2.39 (2.22–2.59) | <0.001 | 1.22 (1.13–1.33) | <0.001 |
| Afrodescendant | 2.39 (2.20–2.60) | <0.001 | 1.10 (1.01–1.19) | 0.025 |
| White | 1.45 (1.28–1.63) | <0.001 | 1.12 (1.01–1.25) | 0.038 |
| **Education** | | | | |
| Higher | Category of comparison | | Category of comparison | |
| High School | 2.81 (2.40–3.29) | <0.001 | 1.87 (1.59–2.20) | <0.001 |
| Primary School | 6.04 (5.22–6.96) | <0.001 | 2.48 (2.11–2.90) | <0.001 |
| None | 11.1 (9.57–12.8) | <0.001 | 3.14 (2.42–4.08) | <0.001 |
| **Region living in** | | | | |
| Lima (capital) | Category of comparison | | Did not enter the model | |
| Coast | 2.47 (2.26–2.70) | <0.001 | Did not enter the model | |
| Mountains | 3.72 (3.42–4.04) | <0.001 | Did not enter the model | |
| Jungle | 4.44 (4.04–4.87) | <0.001 | Did not enter the model | |
| **Type of residence** | | | | |
| Urban | Category of comparison | | Category of comparison | |
| Rural | 3.55 (3.37–3.74) | <0.001 | 1.05 (0.97–1.13) | 0.244 |
| **Marital Status** | | | | |
| Married/cohabiting | Category of comparison | | Did not enter the model | |
| Single | 1.09 (0.97–1.22) | 0.145 | Did not enter the model | |
| Widowed/separated | 1.04 (0.98–1.10) | 0.165 | Did not enter the model | |
| **Physical Limitation** | | | | |
| No | Category of comparison | | Category of comparison | |
| Yes | 1.62 (1.49–1.75) | <0.001 | 1.24 (1.15–1.33) | <0.001 |
| **Wealth Index** | | | | |
| Richest | Category of comparison | | Category of comparison | |
| Rich | 1.68 (1.44–1.96) | <0.001 | 1.22 (1.05–1.43) | 0.012 |
| Middle | 3.22 (2.80–3.70) | <0.001 | 1.91 (1.65–2.21) | <0.001 |
| Poor | 5.62 (4.93–6.40) | <0.001 | 2.55 (2.19–2.96) | <0.001 |
| Poorest | 10.6 (9.39–12.0) | <0.001 | 3.77 (3.22–4.41) | <0.001 |

Crude prevalence ratios (cPR), adjusted prevalence ratios (aPR), 95% confidence intervals (95%: CI) were obtained with Poisson regression with robust variance, with multivariable adjustment by access to education, department of residence and language of birth.

individuals with medium and high educational levels were 3.5 and 5.6 times more likely, respectively, to use dental services compared to those with low educational levels. Similarly, Gaskin et al. [19] reported that elderly individuals with lower educational levels were less likely to have visited a dentist in the last five years and more likely to be edentulous. Additionally, Ju et al. [39] reported that 48% of Australian adults with lower educational levels had not visited a dentist in the last 12 months, and those with higher education were almost 1.5 times more likely to have had a dental visit in the previous year than those with lower education. According to Xu et al. [31], the educational level of older Chinese adults was a determining factor in greater access to oral health services. This was corroborated by Allin and colleagues [40], who argued that the most important factors affecting the use of dental care among older adults globally are predominantly social and cultural in nature, including factors such as language, level of education, and others. Similarly, Drachev and colleagues [41], in Lithuania reported that the use of dental services among older adults was associated with a higher educational level [42].

On the other hand, one of the common characteristics of the deterioration in general and oral health of the elderly is the persistence of chronic, degenerative, and/or intellectual or physical disabilities [43]. For example, spinal cord injuries in older adults cause physical limitations that predispose the individual to constant dependency and emotional distress with catastrophic consequences for their oral health [44]. In this regard, Tiisanoja et al. [45] established that older adults with lower physical capacity had more dental caries. In our study, elderly individuals with physical limitations were associated with lower access to dental care. This could be due to the fact that as the severity of disabilities in older adults increases, general dentists are less predisposed and involved in the care of this type of patient. However, this reality changes when dentists have a specialty or training in geriatric dentistry [46], which is only the case for 40% of dentists in the U.S., for example [47].

Finally, the self-perception of increasing poverty among older adults is a significant factor that hinders the use of dental services and exacerbates the deterioration of their oral health. Not addressing this issue elevates the financial burden on public and private dental health systems [48–50]. In this regard, Kramarow [27] reported that 43% of poor older adults and 43% of near-poor older adults were less likely to have had a dental visit, compared to 74% of non-poor older adults. Similarly, Shahrabani [51] revealed that the decision to undergo dental check-ups among elderly Israelis is affected by cost. Furthermore, Gaskin et al. [19] stated that low-income elderly Americans were less likely to have visited the dentist in the last five years and more likely to be edentulous. Various studies have revealed the relationship between high incomes and a greater likelihood of seeking dental care [25, 36, 39, 52, 53]. Additionally, low per capita family income and unemployment status are significantly associated with never having had a dental appointment [24, 54]. Consequently, elderly individuals with low economic resources will have fewer opportunities to access health services and poorer oral health [55]. This reality will only change if states take a leading role through policies aimed at safeguarding the oral health of older adults [48]. In relation to this, Allen and collaborators [56] asserted that providing financial subsidies for dental care use among the elderly significantly contributes to more regular dental visits in this age group.

Regarding the findings of Azañedo et al. [18] on the factors associated with the use of dental services among older adults in Peru, our results encompass a longer reference period concerning access to dental services. This difference in scope likely explains the variation in findings, with our research identifying a higher use of dental services among older adults compared to their study. However, they agree on determining factors such as educational level, lack of health insurance, and poverty index. Despite oral health being a global priority, Peru does not have a National Oral Health Plan, and the oral health policies of the Peruvian government are

still in their early stages. This state neglect may have influenced the worsening of the oral health of Peruvians and the persistence of oral problems in older adults, as well as the limitations in accessing health services for this population. In this regard, the National Institute of Health (INS), for the first time, has identified oral health research priorities with a validity period from 2022 to 2026, which are expected to guide serious investigations that will serve as evidence for better decision-making in oral health by public management officials, enabling the materialization of the first oral health policies [22].

The difference between the crude and adjusted prevalence rates by type of residence highlights the influence of factors beyond geographic location on the lack of dental service use. While the crude rate indicates a greater lack of use in rural areas, the adjusted rate reveals that much of this disparity can be explained by variables such as educational level, wealth index, and health insurance affiliation, which are unevenly distributed between rural and urban areas. This suggests that economic and educational barriers, rather than location alone, play a critical role in limiting access. Additionally, there may be strong correlations between residence and other factors not accounted for in the model, such as perceived need for care or the availability of dental services in remote areas.

The reduced significance of variables such as ethnoracial self-perception, educational level, type of residence, and wealth index after adjustment likely reflects the interaction and collinearity among these factors. For instance, the impact of wealth index may overlap with educational attainment, as individuals with higher education often possess greater economic resources, facilitating access to dental services. Similarly, the diminished significance of residence type after adjustment could indicate that rural-urban disparities are mediated by socioeconomic factors like education and wealth, rather than being solely attributable to geographic location. These findings suggest that the observed disparities in dental service utilization are not the result of isolated factors but emerge from a complex interplay of overlapping social determinants of health. Furthermore, the exclusion of the geographic region variable from the multivariable model, despite its initial inclusion as an adjustment covariate, was necessary to avoid collinearity issues with other variables that more comprehensively capture structural inequalities. Nonetheless, the role of geographic disparities should not be entirely dismissed, as they remain a relevant factor influencing access to healthcare services in Peru.

The study had limitations inherent to research with secondary data, and there could have been a possible information bias. However, to minimize this bias as much as possible, strict measures were implemented, such as filter/quality control of the data to ensure the quality and accuracy of those data, understanding the importance of verifying the methodology used for data collection, assessing credibility and identifying the accuracy and reliability of the information. In addition, we did not have access to other variables that could have been important, but it is recommended that future research may raise this issue from different perspectives, taking as a basis what has already been found.

Another significant limitation of this study is the inability to differentiate the type of dental service received by participants, as ENDES does not provide details on the nature of dental visits. This limitation makes it impossible to determine whether the visits were preventive, such as dental cleanings, or therapeutic, such as extractions or restorative treatments. This lack of specificity could affect the interpretation of the results, since reported use does not necessarily reflect adequate coverage of participants' dental care needs. For instance, a single visit for a cleaning might be classified as utilization but does not address the needs of individuals requiring more complex and costly procedures, which may be restricted by economic or structural barriers. We acknowledge that this limitation could underestimate the true inequalities in both the utilization and quality of dental services, and we recommend that future research address this gap by including more detailed information on the type and purpose of dental visits.

The exclusion of language as an independent variable in the multivariable model also represents a limitation, which was due to its collinearity with other key covariates such as geographic region and wealth index. These variables more comprehensively capture the structural inequalities influencing dental service use in Peru, leading to their prioritization in the analysis. However, we recognize that language, particularly in contexts where indigenous languages such as Quechua or Aymara are predominant, can pose a significant barrier to accessing healthcare, including dental consultations. Although its specific effect was not independently explored in this study, this highlights the need for future research to delve deeper into the impact of linguistic barriers on access to oral health services, especially in rural areas with high linguistic diversity. Such an approach would provide a better understanding of how the interplay of language, culture, and geography contributes to oral health disparities.

Finally, the selected variables, such as sex, educational level, health insurance, wealth index, and type of residence, are relevant as they represent key sociodemographic factors with a demonstrated impact on access to dental services. Other potentially important variables, such as the specific need for dental care or the reasons for the last dental visit, were not included because they were not available in the ENDES dataset, which limited the scope of the analysis.

## Conclusion

It was reported that 15% of older adults did not utilize dental services, and the determining factors for this outcome were the socio-demographic variables: sex, health insurance coverage, race/ethnicity, presence of physical limitations, educational level, and degree of poverty.

## Author Contributions

**Conceptualization:** Bryan Alexis Cossio-Alva, Miguel Angel Ruiz-Barrueto, Giancarlo Becerra Atoche.

**Data curation:** Rubén Espinoza Rojas, Christian R. Mejia.

**Formal analysis:** Bryan Alexis Cossio-Alva.

**Investigation:** Bryan Alexis Cossio-Alva, Rubén Espinoza Rojas, Miguel Angel Ruiz-Barrueto, Giancarlo Becerra Atoche, Christian R. Mejia, Ibraín Enrique Corrales-Reyes.

**Methodology:** Bryan Alexis Cossio-Alva.

**Project administration:** Bryan Alexis Cossio-Alva.

**Validation:** Bryan Alexis Cossio-Alva, Rubén Espinoza Rojas.

**Visualization:** Bryan Alexis Cossio-Alva, Rubén Espinoza Rojas, Miguel Angel Ruiz-Barrueto, Giancarlo Becerra Atoche, Christian R. Mejia, Ibraín Enrique Corrales-Reyes.

**Writing – original draft:** Bryan Alexis Cossio-Alva, Rubén Espinoza Rojas, Miguel Angel Ruiz-Barrueto, Giancarlo Becerra Atoche, Christian R. Mejia, Ibraín Enrique Corrales-Reyes.

**Writing – review & editing:** Rubén Espinoza Rojas, Miguel Angel Ruiz-Barrueto, Giancarlo Becerra Atoche, Christian R. Mejia, Ibraín Enrique Corrales-Reyes.

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
