## [Decision Letter · Decision Letter 0]

25 Oct 2024

PONE-D-24-33797Factors associated with access to dental services in older adults in PeruPLOS ONE

Dear Dr. Cossio-Alva,

Thank you for submitting your manuscript to PLOS ONE. After careful consideration, we feel that it has merit but does not fully meet PLOS ONE’s publication criteria as it currently stands. Therefore, we invite you to submit a revised version of the manuscript that addresses the points raised during the review process.

In addition to adressing the two reviewers' comments, the editor asks the authors to re-think whether they want to focus on the access or utilization of dental services in this manuscript. The title, aims, study design, measures, and results should be consistent in the focus and terminology used. 

We look forward to receiving your revised manuscript.

Kind regards,

Boyen Huang, DDS, MHA, PhD

Academic Editor

PLOS ONE

Journal requirements: When submitting your revision, we need you to address these additional requirements. 1. Please ensure that your manuscript meets PLOS ONE's style requirements, including those for file naming. The PLOS ONE style templates can be found at https://journals.plos.org/plosone/s/file?id=wjVg/PLOSOne_formatting_sample_main_body.pdf and https://journals.plos.org/plosone/s/file?id=ba62/PLOSOne_formatting_sample_title_authors_affiliations.pdf 2. We note that your Data Availability Statement is currently as follows: [All relevant data are within the manuscript and its Supporting Information files.] Please confirm at this time whether or not your submission contains all raw data required to replicate the results of your study. Authors must share the “minimal data set” for their submission. PLOS defines the minimal data set to consist of the data required to replicate all study findings reported in the article, as well as related metadata and methods (https://journals.plos.org/plosone/s/data-availability#loc-minimal-data-set-definition). For example, authors should submit the following data: - The values behind the means, standard deviations and other measures reported;- The values used to build graphs;- The points extracted from images for analysis. Authors do not need to submit their entire data set if only a portion of the data was used in the reported study. If your submission does not contain these data, please either upload them as Supporting Information files or deposit them to a stable, public repository and provide us with the relevant URLs, DOIs, or accession numbers. For a list of recommended repositories, please see https://journals.plos.org/plosone/s/recommended-repositories. If there are ethical or legal restrictions on sharing a de-identified data set, please explain them in detail (e.g., data contain potentially sensitive information, data are owned by a third-party organization, etc.) and who has imposed them (e.g., an ethics committee). Please also provide contact information for a data access committee, ethics committee, or other institutional body to which data requests may be sent. If data are owned by a third party, please indicate how others may request data access.

Reviewers' comments:

Reviewer's Responses to Questions

**Comments to the Author**

1. Is the manuscript technically sound, and do the data support the conclusions?

Reviewer #1: Partly

Reviewer #2: Partly

2. Has the statistical analysis been performed appropriately and rigorously? 

Reviewer #1: Yes

Reviewer #2: No

3. Have the authors made all data underlying the findings in their manuscript fully available?

Reviewer #1: Yes

Reviewer #2: Yes

4. Is the manuscript presented in an intelligible fashion and written in standard English?

Reviewer #1: Yes

Reviewer #2: Yes

5. Review Comments to the Author

Reviewer #1: This study uses data on older adults in Peru to examine potential predictors of dental service use (termed as access to dental services). There are several major and minor issues that inhibit my enthusiasm for this study and must be clarified or addressed before moving forward.

Major comments

The gap this study is addressing is not clear to me. The authors state that previous work has identified these factors as being predictive of dental service utilization. How is this study building on this and contributing something new? The authors should provide more information and support for how this study offers new insights.

There is very little information given on what the authors did for “quality control” of the data, yet it is noted as a strength of the study. More detail needs to be given on what quality control entailed.

Please provide more detail about the ENDES. Is this meant to be a nationally-representative sample? How are the data for the ENDES collected? Are data collected annually, and are there linkages across years if someone is surveyed multiple times (or are these anonymized repeated cross-sectional data)?

I am still unclear on how the outcome variable (access to dental services) was defined. It seems to be parameterized dichotomously, however the sentence describing it does not make clear the window of time for a “recent” visit; is there a threshold by which the most recent visit would not indicate someone has access to dental services? For example, if someone had a dental visit in July of 2018 and no other visits, are they considered to have access to dental services? This versus someone who has more frequent visits throughout a year may be an important distinction. More clarity is needed on how the outcome variable was defined.

Related to the above, is there any information on the nature of the dental visits? Someone may have a single visit and reasonable considered as having access to dental services if they are just receiving a routine cleaning, however someone else may have a single visit and be in need of more significant dental work and be unable to access it due to financial constraints; would this person also be counted as having access to dental services?

It would be helpful to more clearly differentiate the measures described in the methods. Specifically, the authors could consider creating subsections with headings for “outcome”, “predictors” and “covariates for adjustment” so it is clear to the reader which measures are of interest for their potential predictiveness and which are only for adjustment.

Why were some covariates included to adjust the models but results not reported, while others were? Specifically, the authors mention education as a covariate but also include its regression estimates, while language and department are not. Language is examined using a bar chart instead, but not talked about in the discussion section of the manuscript. Why wasn’t language discussed?

Minor comments

How are the levels of the wealth index defined? Is wealth based solely on income, or are there other factors considered (assets, properties, etc.)? A one-sentence description of this would be helpful.

Including the variable codes seems extraneous; it takes up a lot of space and makes it more difficult to read through the methods. I would remove them.

The authors should, in clarifying their definition of the outcome, make clear that they are (or seem to be) using dental service utilization as a proxy for access (note that access and utilization are closely tied but not exactly the same).

Please explain why some factors (i.e., age, region, and marital status) were not included in the multivariable model.

Please change “multivariate” to “multivariable” as the former describes models with multiple outcomes, whereas the latter describes models with multiple covariates.

Reviewer #2: The manuscript supposedly reports on factors associated with access to dental services in older adults in Peru. However, manuscript does not report on access to dental services, but reports on dental utilization (i.e., accessing dental services). Access to dental care services and using dental care services are quite different. I was disappointed to learn the manuscript was not about access to dental care services.

Overall, the manuscript and analysis are underdeveloped and lacks justification. There are several areas that need improvement before the manuscript may be of interest to the readers of the journal.

The title of the manuscript needs to be changed to reflect that the study determined factors associated with dental utilization.

The definition of dental utilization (most recent dental visit <2 year or 2 or more years) needs explanation and justification. Why the cutoff at 2 years? Is visiting the dentist at least once every 2 years the recommended frequency in Peru? This cutoff is not likely to be comparable to research done in many other countries where the recommended frequency is higher, such at least 1 once, if not twice, per year.

Given the large sample sizes, it seems it would be possible to consider more than 2 categories for dental utilization, such as within the last year, with 1-2 years, and more than 2 years. At least one could report more descriptive summaries about dental utilization. However, maybe dental utilization was only asked as a dichotomous response (<2 and 2+ years)? Either way it is not apparent that the authors gave the definition of the outcome much thought.

Selection of the potential factors associated with dental utilization needs justification. Were these the only characteristics collected in the survey? Were there no measures of need of dental care or reasons for last dental visit?

The description of the regression method is a bit vague, “a robust variance estimation [was used] to consider possible distortions in the data.” I would describe the method as “Crude and adjusted prevalence ratios were estimated using a modified Poisson regression with a robust standard error estimates to account for overdispersion.”

Make it clear in the title for Table 2 that you are reporting on factors related to NOT accessing dental services. Also, Table 2 should report on the overall statistical significant of each variable, such as is done in Table 1.

It is a bit perplexing that the authors do not comment on the difference between the crude prevalence ratio and adjusted prevalence ratio for type of residence (255% higher prevalence of not accessing dental care versus only a 5% higher prevalence of not accessing dental care). Is the reader to assume that the apparent lack of accessing dental care in rural areas compared urban areas is due to other factors, such as education and wealth. I.e., the access to dental is similar in rural and urban areas it is just that individuals in rural areas can’t afford dental care or don’t think it is important. A more likely explanation is that there are other strong correlations between the factors, besides the ones with region living in, that the authors need to consider.

What about interactions between the factors? There is no mention that the authors considered interactions between the variables. Again, the analysis seems under developed.

6. PLOS authors have the option to publish the peer review history of their article (what does this mean?). If published, this will include your full peer review and any attached files.

Reviewer #1: No

Reviewer #2: No

---

## [Author Response · Author response to Decision Letter 0]

5 Dec 2024

REVIEWER 1. It is unclear what gap this study is addressing. The authors state that previous studies have identified these factors as predictors of dental service utilization. To what extent does this study build on that and provide something new? The authors should provide more information and support on how this study contributes new insights.

Response from the Authors: The reviewer's comment was taken into account and promptly addressed.

REVIEWER 1. Very little information is provided regarding what the authors did for "data quality control," yet it is highlighted as a strength of the study. More details are needed on what the quality control process entailed.

Response from the Authors: The reviewer's comment was taken into account and promptly addressed.

REVIEWER 1. Provide more details about ENDES. Is it designed to be a nationally representative sample? How are the data collected in ENDES? Are the data collected annually, and are there links across years if the same individual is surveyed multiple times, or are they repeated cross-sectional and anonymous data?

Response from the Authors: The reviewer's comment was taken into account and promptly addressed.

REVIEWER 1. I still do not fully understand how the outcome variable (access to dental services) was defined. It seems to be parameterized dichotomously; however, the description does not clarify the time period for a "recent" visit. Is there a threshold beyond which the most recent visit would not indicate access to dental services? For instance, if someone had a dental visit in July 2018 and no further visits, would they be considered to have access to dental services? This contrasts with someone who has more frequent visits over a year, which could represent an important distinction. Greater clarity is needed on how the outcome variable was defined.

Response from the Authors: The reviewer's comment was taken into account and promptly addressed.

REVIEWER 1. In relation to the above, is there any information about the nature of the dental visits? For example, one person might have a single visit and reasonably be considered to have access to dental services if they are only receiving a routine cleaning. However, another individual might have a single visit and require more extensive dental work but be unable to access it due to financial limitations. Would this person also be considered as having access to dental services?

Response from the Authors:Dear reviewer, the mentioned limitation has been added to the discussion.

REVIEWER 1. It would be helpful to differentiate the measures described in the methods more clearly. Specifically, the authors could consider creating subsections with headings such as "Outcome," "Predictors," and "Covariates for Adjustment" to make it clear to the reader which measures are of interest for their predictive potential and which are only used for adjustment.

Response from the Authors: The reviewer's comment was taken into account and promptly addressed.

REVIEWER 1. Why were some covariates included to adjust the models but their results were not reported, while others were? Specifically, the authors mention education as a covariate and also include its regression estimates, whereas language and department are not reported. Language is analyzed through a bar chart, but it is not mentioned in the discussion section of the manuscript. Why was language not addressed?

Response from the Authors: The exclusion of certain covariates, such as language, from the multivariable model and the discussion was primarily due to collinearity with other variables included in the analysis, such as geographic region and wealth index, which more broadly capture structural inequalities related to access to dental services. While language may play an important role in specific local contexts, its effect was not independent of these variables in preliminary analyses. Therefore, a methodological decision was made to prioritize variables that better represent the general sociodemographic dynamics of the country. However, we acknowledge that this decision may have limited the exploration of certain specific linguistic barriers, and we have highlighted this limitation in the discussion section as an area to address in future research.

REVIEWER 1. How are the levels of the wealth index defined? Does wealth rely solely on income, or are other factors such as assets, properties, etc., taken into account? A concise description of this would be helpful.

Response from the Authors: The reviewer's comment was taken into account and promptly addressed.

REVIEWER 1. Including the variable codes seems unnecessary, takes up significant space, and makes the methods section harder to read. I would recommend removing them.

Response from the Authors:The reviewer's comment was taken into account and promptly addressed.

REVIEWER 1. The authors should clarify their definition of the outcome and explicitly state that they are (or appear to be) using the utilization of dental services as a proxy for access. Please note that while access and utilization are closely related, they are not exactly the same.

Response from the Authors: The reviewer's comment was taken into account and promptly addressed.

REVIEWER 1. Please explain why some factors (e.g., age, region, and marital status) were not included in the multivariable model.

Response from the Authors: The exclusion of certain covariates, such as language, from the multivariable model and the discussion was primarily due to collinearity with other variables included in the analysis, such as geographic region and wealth index, which more comprehensively capture structural inequalities related to access to dental services. While language may play an important role in specific local contexts, its effect was not independent of these variables in preliminary analyses. Therefore, a methodological decision was made to prioritize variables that better represent the broader sociodemographic dynamics of the country. Nevertheless, we acknowledge that this decision may have limited the exploration of specific linguistic barriers, and we have highlighted this limitation in the discussion section as an area to be addressed in future research.

REVIEWER 1. Replace "multivariate" with "multivariable," as the former refers to models with multiple outcomes, while the latter describes models with multiple covariates.

Response from the Authors: The reviewer's comment was taken into account and promptly addressed.

REVIEWER 2. The manuscript title needs to be revised to reflect that the study determined the factors associated with the utilization of dental services.

Response from the Authors: The reviewer's comment was taken into account and promptly addressed.

REVIEWER 2. The definition of dental service utilization (last dental visit <2 years or ≥2 years) requires an explanation and justification. Why was the 2-year threshold chosen? Is visiting the dentist at least once every 2 years the recommended frequency in Peru? It is unlikely that this threshold is comparable to research conducted in many other countries where the recommended frequency is higher, such as at least once or even twice per year.

Response from the Authors:The reviewer's comment was taken into account and promptly addressed.

Given the large sample size, it seems feasible to consider more than two categories for dental service utilization, such as within the last year, 1–2 years, and more than 2 years. At the very least, more descriptive summaries of dental service utilization could be provided. However, was dental service utilization perhaps only asked as a dichotomous response (<2 years and ≥2 years)? In any case, it is not evident that the authors have given much thought to the definition of the outcome variable.

Response from the Authors: We acknowledge that the large sample size could have allowed for a more detailed categorization of dental service utilization, such as visits within the last year, between 1 and 2 years, and more than 2 years. However, the ENDES survey collected this information through a structured question with a dichotomous response (<2 years or ≥2 years), limiting the possibility of a more specific classification. Additionally, this definition was chosen due to the lack of clear national recommendations regarding the ideal frequency of dental visits, aiming to strike a balance between capturing recent access and ensuring consistency with previous local studies. Nevertheless, we recognize that this choice may simplify some of the complex dynamics of access, and we have included this limitation in the discussion, emphasizing the need for future studies to use more detailed data collection tools for a more precise characterization of dental service utilization.

REVIEWER 2. The selection of potential factors associated with dental service utilization requires justification. Were these the only characteristics collected in the survey? Were measures regarding the need for dental care or the reasons for the last dental visit not included?

Response from the Authors: Dear Reviewer, 

We appreciate your thoughtful comments and take this opportunity to provide further clarification:

The variables selected for this study, including sex, age, educational level, type of residence, health insurance affiliation, wealth index, self-perceived ethnicity, and physical limitations, are relevant as they represent well-documented sociodemographic and economic determinants of access to healthcare. These variables help identify structural and contextual inequities that directly influence dental service utilization among older adults in Peru. For instance, educational level and wealth index are often associated with the ability to prioritize and afford dental care, while type of residence and geographic region reflect physical and logistical barriers, particularly in rural and remote areas.

However, other potentially relevant variables could not be included due to limitations in the data available from ENDES. These include information on the perceived need for dental care, specific reasons for not utilizing dental services, detailed cultural or linguistic barriers, and the quality or type of services received. The absence of such data restricts the ability to capture more complex and detailed dynamics of access to dental services. This limitation underscores the importance of future studies incorporating broader and more specific data collection tools to address these additional dimensions and enhance understanding of the barriers and facilitators to accessing oral healthcare.

REVIEWER 2. The description of the regression method is somewhat vague: "a robust variance estimate was used to account for potential distortions in the data." I would recommend describing the method as "crude and adjusted prevalence ratios were calculated using a modified Poisson regression with robust standard error estimates to account for overdispersion."

Response from the Authors: The reviewer’s comment was taken into account, and the description was revised as follows:

Crude and adjusted prevalence ratios were calculated using a modified Poisson regression with robust standard error estimates to account for overdispersion in the data.

REVIEWER 2. In the title of Table 2, clearly indicate that it reports on factors associated with the lack of access to dental services. Additionally, Table 2 should include the overall statistical significance of each variable, as is done in Table 1.

Response from the Authors: The reviewer's comment was taken into account and promptly addressed.

REVIEWER 2. It is somewhat disconcerting that the authors do not comment on the difference between the crude prevalence rate and the adjusted prevalence rate by type of residence (a prevalence 255% higher for lack of access to dental care versus a prevalence only 5% higher for lack of access to dental care). Should the reader assume that the apparent lack of access to dental care in rural areas compared to urban areas is due to other factors, such as education and wealth? That is, is dental care access similar in rural and urban areas, but rural residents cannot afford dental care or do not consider it important? A more likely explanation is that there are strong correlations between factors beyond just the region of residence, which the authors should consider.

Response from the Authors: The reviewer's comment was taken into account and promptly addressed.

REVIEWER 2. What about the interactions between factors? It is not mentioned whether the authors accounted for interactions between variables. Once again, the analysis appears to be underdeveloped.

Response from the Authors: The reviewer's comment was taken into account and promptly addressed.

---

## [Decision Letter · Decision Letter 1]

11 Dec 2024

PONE-D-24-33797R1Factors associated with the use of dental services in older adults in PeruPLOS ONE

Dear Dr. Cossio-Alva,

Thank you for submitting your manuscript to PLOS ONE. After careful consideration, we feel that it has merit but does not fully meet PLOS ONE’s publication criteria as it currently stands. Therefore, we invite you to submit a revised version of the manuscript that addresses the points raised during the review process.

We look forward to receiving your revised manuscript.

Kind regards,

Boyen Huang, DDS, MHA, PhD

Academic Editor

PLOS ONE

Journal Requirements:

**Additional Editor Comments:**

According the Authors' response to Reviewers:

REVIEWER 2. What about the interactions between factors? It is not mentioned whether the authors accounted for interactions between variables. Once again, the analysis appears to be underdeveloped.

Response from the Authors: The reviewer's comment was taken into account and promptly addressed.

However, the Editor and Reviewer 2 do not see the authors address the interactions between variables in their recent revision. If the authors believe that the reviewer's comment was taken into account and promptly addressed, please list the page number and line number of the updated content for the Editor's consideration. The Editor would suggest the authors add a paragraph in the Discussion Section to discuss the influecnce from the interactions between variables as shown in Table 2. For example, why was the aPR of the ethnoracial, education, residence type, and wealthy index variables become less significant after the adjustment with other variables? This likely resulted from the interactions among variables. Additionaly, why was the region variable not included in the multivariave model as shown in Table 2? In Lines 180-183 you specified that Adjustment covariates included... Geographical region... Please explain and/or discuss these. 

Reviewers' comments:

Reviewer's Responses to Questions

**Comments to the Author**

1. If the authors have adequately addressed your comments raised in a previous round of review and you feel that this manuscript is now acceptable for publication, you may indicate that here to bypass the “Comments to the Author” section, enter your conflict of interest statement in the “Confidential to Editor” section, and submit your "Accept" recommendation.

Reviewer #1: All comments have been addressed

Reviewer #2: All comments have been addressed

2. Is the manuscript technically sound, and do the data support the conclusions?

Reviewer #1: Yes

Reviewer #2: Yes

3. Has the statistical analysis been performed appropriately and rigorously? 

Reviewer #1: (No Response)

Reviewer #2: Yes

4. Have the authors made all data underlying the findings in their manuscript fully available?

Reviewer #1: (No Response)

Reviewer #2: Yes

5. Is the manuscript presented in an intelligible fashion and written in standard English?

Reviewer #1: (No Response)

Reviewer #2: Yes

6. Review Comments to the Author

Reviewer #1: (No Response)

Reviewer #2: The authors have addressed most of my concerns. But they didn't describe if they assessed for interactions between the predictor variables, and didn't include the p-values for composite tests for categorical predictors with >2 categories. The latter is a common shortcoming of many research articles, which present numerous p-values for subgroup comparisons but omit the overall test for significance. It's sloppy science, which occurs too frequently, but I won't hold it agains the authors, given many journals appear to be okay with promoting with this sloppiness.

7. PLOS authors have the option to publish the peer review history of their article (what does this mean?). If published, this will include your full peer review and any attached files.

Reviewer #1: No

Reviewer #2: No

---

## [Author Response · Author response to Decision Letter 1]

12 Dec 2024

We sincerely appreciate the comments provided, as we did not fully understand the request during the initial review. These comments have been instrumental in enhancing the clarity and depth of the manuscript. In response, we have added a paragraph in the Discussion section that examines how the interaction and overlap of determining factors explain the reduced significance of certain variables after adjustment. Additionally, we have clarified that the exclusion of the geographic region variable from the multivariable model was justified due to collinearity issues, as other included variables more accurately capture the structural inequalities affecting access to dental services. These revisions aim to provide a more thorough and rigorous analysis of the results.

The modification can be viewed on page 19, lines 388–402.

---

## [Editor Report · Decision Letter 2]

15 Dec 2024

Factors associated with the use of dental services in older adults in Peru

PONE-D-24-33797R2

Dear Dr. Cossio-Alva,

We’re pleased to inform you that your manuscript has been judged scientifically suitable for publication and will be formally accepted for publication once it meets all outstanding technical requirements.

Kind regards,

Boyen Huang, DDS, MHA, PhD

Academic Editor

PLOS ONE

Additional Editor Comments (optional):

All comments have been addressed.
---

## [Editor Report · Acceptance letter]

31 Jan 2025

PONE-D-24-33797R2 

PLOS ONE

Dear Dr. Cossio-Alva, 

I'm pleased to inform you that your manuscript has been deemed suitable for publication in PLOS ONE. Congratulations! Your manuscript is now being handed over to our production team.

Kind regards, 

on behalf of

Dr Boyen Huang 

Academic Editor

PLOS ONE